# Are objective measures of physical capability related to accelerated epigenetic age? Findings from a British birth cohort

Andrew J Simpkin,[1,11] Rachel Cooper,[2] Laura D Howe,[1] Caroline L Relton,[1] George Davey Smith,[1] Andrew Teschendorff,[3,4,5] Martin Widschwendter,[3] Andrew Wong,[2] Diana Kuh,[2] Rebecca Hardy[2]

For numbered affiliations see end of article.

**Correspondence to**
Dr Andrew J Simpkin;
Andrew.Simpkin@bristol.ac.uk

## ABSTRACT

**Objectives** Our aim was to investigate the association of epigenetic age and physical capability in later life. Having a higher epigenetic than chronological age (known as age acceleration (AA)) has been found to be associated with an increased rate of mortality. Similarly, physical capability has been proposed as a marker of ageing due to its consistent associations with mortality.

**Setting** The MRC National Survey of Health and Development (NSHD) cohort study.

**Participants** We used data from 790 women from the NSHD who had DNA methylation data available.

**Design** Epigenetic age was calculated using buccal cell (n=790) and matched blood tissue (n=152) from 790 female NSHD participants. We investigated the association of AA at age 53 with changes in physical capability in women from ages 53 to 60–64. Regression models of change in each measure of physical capability on AA were conducted. Secondary analysis focused on the relationship between AA and smoking, alcohol, body mass index (BMI) and socioeconomic position.

**Outcome measures** Three objective measures of physical capability were used: grip strength, standing balance time and chair rise speed.

**Results** Epigenetic age was lower than chronological age (mean 53.4) for both blood (50.3) and buccal cells (42.8). AA from blood was associated with a greater decrease in grip strength from ages 53 to 60–64 (0.42 kg decrease per year of AA, 95% CI 0.03, 0.82 kg; p=0.03, n=152), but no associations were observed with standing balance time or chair rise speed. Current smoking and lower BMI were associated with lower epigenetic age from buccal cells.

**Conclusions** We found evidence that AA in blood is associated with a greater decrease in grip strength in British females aged between 53 and 60–64, but no association with standing balance time or chair rise speed was found.

## INTRODUCTION

There has been considerable recent interest in epigenetic biomarkers of ageing,[1–6] which use an individual's DNA methylation data to estimate their 'epigenetic age', a concept

---

## Strengths and limitations of this study

► Our study is one of the first to examine epigenetic age from different tissues on the same individuals in relation to objective measures of physical capability, which are key markers of healthy ageing.

► We used serial measures of physical capability on the same individuals over time, allowing for better inferences on changes in physical capability in late midlife, compared with having cross sectional data.

► A limitation of our findings is the lack of generalisability - the subsample of the cohort consisted of females only with repeated measures at ages 53 and 60-64 and was restricted to those with complete information on particular variables of interest (i.e. blood and buccal samples).

---

that could be considered a form of biological age. The Horvath age estimation method[1] found a correlation of 0.96 between chronological and epigenetic age, with individual estimates of epigenetic age within 3 years of chronological age on average. Epigenetic age has the potential to assess our biological age, but little is known about its relationship with our basic physiology. Moreover, several recent papers have found that the difference between epigenetic and chronological age (known as age acceleration, denoted AA) has biological significance. A positive AA indicates an individual's epigenetic age is ahead of their chronological age, a negative AA (ie, age deceleration) suggests an individual has younger epigenetic age than chronological age. For example, positive AA has been found to be associated with obesity,[7] Down's syndrome,[8] HIV,[9] menopause[10] and all-cause mortality.[11 12]

Lower physical capability, assessed using objective measures such as grip strength, chair rise speed and standing balance time

have been found to be associated with all-cause mortality.[13] These findings, established through a systematic review of mainly older populations, were also observed using data on physical capability in midlife from the MRC National Survey of Health and Development (NSHD),[14–16] which has followed 5362 individuals born in the same week of March 1946.

It is pertinent to understand the mechanisms underpinning the association between epigenetic age and mortality, since epigenetics are a potentially modifiable risk factor. A recent study identified a cross-sectional association between epigenetic AA and lower grip strength in an older population using data from the Lothian Birth Cohort 1936.[17] There was no strong evidence for links between epigenetic AA and changes in grip strength, lung function or cognition from age 70 to 76. More recent studies have reported no evidence between epigenetic age and either cognitive[18] or composite measures of biomarker[19] of ageing. In the present article, we sought to investigate the associations between epigenetic AA at age 53 and objective measures of physical capability at ages 53 and 60–64 in the NSHD. We hypothesised that individuals with positive AA (ie, with epigenetic age higher than expected based on a linear regression of epigenetic age on chronological age) would have lower average physical capability scores and greater declines than those with lower epigenetic age. We also use data from the NSHD to investigate whether increased epigenetic age is associated with mortality risk factors; smoking,[20] higher body mass index (BMI)[21 22] and more disadvantaged socioeconomic position (SEP).[23]

## METHODS
### Study participants

DNA was first collected in NSHD participants at age 53[24] in 1999; following quality control (QC), the study sample with epigenetic information available consisted of 790 women who had a buccal cell sample taken at age 53 (mean age 53.4, SD 0.16, range 53–54), and who had complete information on epidemiological variables of interest and follow-up. Among these 790 women, there were 152 who also had epigenetic information available from blood at age 53. These 152 were originally selected for a case-control study of cancer, and consisted of 75 incident cancer cases after age 53 and 77 controls randomly selected from those with complete data available for the cancer study.[16 25 26] Mortality risk factor data were available at age 53 on smoking status (current, never or ex-smoker), nurse measured height (cm) and weight (kg) (used to calculate BMI (kg/m$^2$)). Childhood SEP was indicated by father's occupational class and SEP in adulthood by own occupational social class at 53 and educational qualifications by age 26. Father's occupational class and own occupational class in adulthood were each defined according to the Registrar General's social classification: unskilled, partly skilled, skilled (manual), skilled (non-manual), intermediate or professional. Education level attained was classified as none, vocational, sub-General Certificate of Secondary Education (GCSE), O level, A level, degree or higher.

### Study outcomes

The three measures of physical capability were grip strength (kg), standing balance time (s) and chair rise time (s) measured at age 53 and again at age 60–64 by nurses using standardised protocols.[27] Grip strength was ascertained isometrically using an electronic dynamometer which was calibrated using a back-loading rig and was stable to within 0.5 kg. Two values from each hand were recorded at 53, and three in each hand at 60–64, with the maximum of the first four values at each age used for analysis. The standing balance test recorded the times that participants could stand on one leg up to a maximum of 30 s first with eyes open and then repeated with eyes closed. Balance times with eyes closed were used for analysis and these were log transformed to reduce skewness. Chair rise time was measured using a stopwatch and recorded as the time taken to rise from a seated position to a standing position with a straight back and legs followed by a return to a seated position as fast as possible, repeated 10 times. Chair rise speed was then calculated by dividing the number of rises (ie, 10) by the time taken in minutes. This was done to make high scores correspond to good performance, as for the other two measures. Nurses recorded whether the participant was unwilling or unable to perform each of the tests along with the reason for this.

Composite capability scores were generated by combining performance on grip strength, balance time and chair rise speed using methods previously described.[14] In brief, each measure was rescaled to a 0 (low) to 1 (high) scale before aggregation into a composite score from 0 to 3 at ages 53 and 60–64. Standing balance time was rescaled by dividing by 30 s (the maximum time allowed); height adjusted grip strength and chair rise time were rescaled by dividing by the 99th percentile. Those unable to carry out a test for health reasons were assigned a score of 0 for that test.

### DNA methylation data

DNA methylation was measured using the Infinium HumanMethylation450 BeadChip (Illumina) in NSHD participants who had biological samples collected in 1999; 638 (buccal cell only) and 152 (buccal cell and blood).[25] QC and normalisation was performed on each of the 790 buccal samples and then separately on the 152 matched whole blood samples. For each, the minfi package was used to process raw idat data files,[28] using the Illumina definition of beta-values and extracting p values of detection for each sample. The Illumina methylation beta-value of a given CpG site is found from the intensity of the methylated (M) and unmethylated (U) alleles, as the ratio of fluorescent signals $\beta = Max(M,0)/[Max(M,0)+Max(U,0)+100]$. The level of methylation is expressed as a 'beta' value (β-value), ranging from 0 (no

cytosine methylation) to 1 (complete cytosine methylation). As a further QC step, probes that contained <95% of signals detectable above background signal (detection p<0.01) were removed from further analysis, and the rest of missing values were imputed using the k-nearest neighbours imputation procedure.[29] To correct for the well-known bias of type 2 probes, we used the subset-quantile within normal array package.[30] To check robustness of this correction procedure, we verified that results were largely unchanged using beta mixture quantile normalisation.[31] This completed the intrasample normalisation. All participants provided written informed consent. The Central Manchester Ethics Committee approved the use of these samples for epigenetic studies of health.

## Epigenetic age

Using the online epigenetic clock calculator (http://labs. genetics.ucla.edu/horvath/dnamage), we obtained DNA methylation estimated age using the Horvath[1] method. The raw DNA methylation β-values were generated from the 152 blood and 790 buccal cell samples. Along with epigenetic age, the online calculator estimates raw AA differences (epigenetic minus chronological age) and AA residuals (the residuals from a linear regression of epigenetic age on chronological age). Our main exposure of interest is the latter AA residual, which we will call AA. AA values from blood were corrected for estimated cell type heterogeneity using the Houseman method.[32] The Houseman estimated cell counts were included in the regression of epigenetic age on chronological age to get cell count adjusted AA.

## Statistical analysis

Median absolute error was used to investigate the relationship between chronological age and epigenetic age from blood and buccal tissue, with correlation being secondary given the low range of actual age in NSHD. Changes in each physical capability measure were considered the main outcomes, with the differences in grip strength, chair rise speed, balance time and composite score from age 53 to age 60–64 being used for analysis. Using this unconditional change model allows us to directly compare our results with those from the Lothian Birth Cohort 1936[17]. The differences were regressed on AA from blood and buccal tissue separately. We fitted unadjusted regression models followed by models adjusted for age, height and BMI and then additionally adjusted for smoking and both childhood and adult SEP. Linear regression was also used to test the association of AA (from both blood and buccal cells) at age 53 with each physical capability measure and the composite score at both ages 53 and 60–64. As a secondary analysis with AA as the outcome, we carried out unadjusted regression analysis of the known mortality risk factors of height, BMI, smoking and SEP (both childhood and adult).

In a sensitivity analysis, we reran the main models with inclusion of those women who were unable to perform each of the three physical capability tests for health reasons (table 1 includes percentage unable to perform each task). To enable their inclusion, women who were unable to complete a test for health reasons were allocated the minimum value observed at either age.

## Replication

Findings were tested for replication using cross-sectional data from the mothers of the Avon Longitudinal Study of Parents and Children (ALSPAC).[33 34] ALSPAC recruited 14 541 pregnant women with expected delivery dates between April 1991 and December 1992. Of these initial pregnancies, there were 14 062 live births and 13 988 children who were alive at 1 year of age. The study website contains details of all the data that are available through a fully searchable data dictionary (http://www.bris.ac. uk/alspac/researchers/data-access/data-dictionary). DNA methylation and epigenetic age were available from 988 ALSPAC mothers at mean age of 46.9 (SD 4.7 years, range 31–60) as part of the Accessible Resource for Integrated Epigenetics Studies (ARIES) project.[35] The ARIES study is a subsample of ALSPAC, which generated DNA methylation for 1000 families who had biological samples available at each of five time points: umbilical cord blood at birth, peripheral blood at age 7 and 17 in children and peripheral blood during pregnancy and at 18 years follow-up for mothers. The 1000 families were randomly selected from those who had full data available. We used the mother's follow-up data to replicate our analysis because they best reflected the available NSHD women. All DNA methylation wet-lab and preprocessing analyses were performed at the University of Bristol as part of the ARIES project. Following extraction, DNA was bisulphite converted using the Zymo EZ DNA MethylationTM kit (Zymo, Irvine, California, USA). Infinium HumanMethylation450 BeadChips were used to measure genome-wide DNA methylation levels at over 485 000 CpG sites. The arrays were scanned using an Illumina iScan, with initial quality review using GenomeStudio. The assay detects methylation of cytosine at CpG islands using two site-specific probes—one to detect the methylated (M) locus and one to detect the unmethylated (U) locus. Single-base extension of the probes incorporates a labelled chain-terminating ddNTP, which is then stained with a fluorescence reagent. The ratio of fluorescent signals from the methylated site versus the unmethylated site determines the level of methylation at the locus. The level of methylation is expressed as a β-value, ranging from 0 (no cytosine methylation) to 1 (complete cytosine methylation). β-Values are reported as percentages.

Grip strength, balance time and chair rise speed along with height, BMI, smoking and SEP (adulthood only) were available from these same women. Grip strength was assessed using the Jamar handgrip dynamometer and was recorded to the nearest 1 kg using both the right and left hands. Two measures were taken in each hand and the maximum of these values was used. In the chair rise test, the participant was asked to rise from a sitting position to a straight-legged fully standing position five times while

**Table 1** Descriptive statistics for continuous variables

| Continuous variable | Age when measured | Mean (SD) | N | N unable (%) |
|---|---|---|---|---|
| Age (years) | 53 | 53.44 (0.16) | 790 | – |
| | 64 | 63.09 (1.09) | 623 | – |
| Epigenetic age from buccal (years) | 53 | 42.83 (5.71) | 790 | – |
| Age acceleration from buccal (years) | 53 | 0.00 (5.66) | 790 | – |
| Epigenetic age from blood (years) | 53 | 50.28 (4.34) | 152 | – |
| Age acceleration from blood (years) | 53 | 0.00 (4.34) | 152 | – |
| Height (cm) | 53 | 161.43 (5.61) | 790 | – |
| BMI (kg/m$^2$) | 53 | 28.13 (6.43) | 784 | – |
| | 64 | 29.10 (7.50) | 622 | – |
| Grip strength (kg) | 53 | 28.18 (8.15) | 767 | 18 (2) |
| | 64 | 25.71 (7.94) | 575 | 21 (3) |
| Grip strength change (kg) | 64 | −2.62 (8.53) | 560 | |
| Chair rise speed (stands/min) | 53 | 30.98 (9.56) | 737 | 40 (5) |
| | 64 | 24.54 (7.82) | 577 | 40 (5) |
| Chair rise speed change (stands/min) | 64 | −6.87 (10.07) | 556 | |
| Balance time (log-seconds) | 53 | 1.77 (0.60) | 734 | 32 (8) |
| | 64 | 1.52 (0.55) | 593 | 24 (3) |
| Balance time change (log-seconds) | 64 | −0.27 (0.68) | 561 | – |
| Composite score | 53 | 1.31 (0.33) | 750 | – |
| | 64 | 1.19 (0.37) | 591 | – |
| Composite score change | 64 | −0.13 (0.36) | 558 | – |

| Categorical variable | Age when measured | Category | N (%) | |
|---|---|---|---|---|
| Smoking | 53 | Never | 384 (49) | |
| | | Ex | 239 (30) | |
| | | Current | 167 (21) | |
| Childhood SEP | 53 | Professional | 57 (7) | |
| | | Intermediate | 160 (21) | |
| | | Skilled (non-manual) | 123 (16) | |
| | | Skilled (manual) | 245 (31) | |
| | | Partly skilled | 148 (19) | |
| | | Unskilled | 48 (6) | |
| Adult SEP | 53 | Professional | 14 (2) | |
| | | Intermediate | 261 (33) | |
| | | Skilled (non-manual) | 286 (36) | |
| | | Skilled (manual) | 57 (7) | |
| | | Partly skilled | 119 (15) | |
| | | Unskilled | 50 (7) | |
| Education | 53 | None | 275 (35) | |
| | | Vocational | 44 (5) | |
| | | Sub-GCE | 38 (5) | |
| | | O level | 201 (26) | |
| | | A level | 106 (14) | |
| | | Burham A2 | 75 (10) | |
| | | Degree | 32 (4) | |
| | | Postgraduate | 2 (1) | |

BMI, body mass index; GCE, General Certificate of Education; SEP, socioeconomic position.

being timed. Chair rise speed was then calculated by dividing five by the total time required. This differs from NSHD in having five total stands, although most of the between-study variability would be resolved by using chair rise speed rather than total time taken. In the balance time test, the participant stood next to a table and asked to choose a leg and raise it off the floor to ankle height. The participant was timed until they lost their balance and dropped their foot or had to reach out to the table for support. If the participant remained on one leg for longer than 30 s, they were stopped. The process was repeated with eyes closed, which was used for analysis to mirror the NSHD measure.

## RESULTS

The descriptive statistics are displayed in table 1. The average epigenetic age was 42.8 (SD 5.71 years) using DNA methylation from buccal tissue and 50.3 (SD 4.34 years) using DNA methylation from blood. These both underestimate the average chronological age of 53.4 years (SD 0.16 years). The median absolute error between chronological and epigenetic age is 10.5 and 4.1 years for buccal and blood, respectively. Correlation with chronological age was much lower than previously reported: 0.022 (p=0.79) for blood age and 0.115 (p=0.16) for buccal age, but this correlation is a less appropriate assessment due to the narrow age range (SD of age=0.16 years). Correlation was slightly higher between the two epigenetic ages, with a Pearson correlation coefficient of 0.190 (p=0.02).

AA, being the residual of a regression of epigenetic age on chronological age, has a mean close to zero by definition. However, the variance and range are larger for AA of buccal tissue compared with blood tissue. Average levels of each physical capability measure changed in the expected direction, with a mean decrease in grip strength of 2.6 kg (SD 8.5 kg), chair rise speed of 6.8 stands/min (SD 10.1 stands/min) and balance time of 0.27 log-seconds (SD 0.68 log-seconds) from age 53 to 60–64. This is reflected in an average decrease of 0.12 units in the composite score for physical capability.

### Age acceleration and physical capability
#### Change in physical capability
For a 1 year increase in AA, grip strength decreased by an additional 0.42 kg (95 % CI 0.03 to 0.82 kg; p=0.03) from age 53 to 60–64 after adjusting for height, BMI, education and SEP (table 2). There was no strong evidence for an association between AA and change in chair rise speed (0.06 higher stands/min per 1 year AA 95 % CI −0.40 to 0.52 stands/min; p=0.80) or balance time (0.01 log-seconds lower per 1 year AA, 95 % CI −0.04 to 0.02 log-seconds; p=0.55). The weak associations of AA from buccal cells and physical capability were in the expected direction.

### Separate analysis of physical capability measured at 53 and 60–64
There were no associations between physical capability at age 53 and epigenetic AA, either from blood or buccal samples (online supplementary table S1). Effect sizes were much smaller in buccal AA compared with blood AA for grip strength and particularly for chair rise speed. The associations of AA with grip strength, balance time and the composite score were positive, indicating greater AA was associated with better performance, that is, the opposite direction to that expected. Similarly, there was little evidence for an association between epigenetic age and any of the physical capability markers or the composite score at age 60–64 (online supplementary table S2). Here, the effect of AA was in the expected negative direction for grip strength, chair rise speed and balance time, with stronger effects observed from blood AA than buccal AA.

### Age acceleration and mortality risk factors
There were positive associations between BMI at age 53 and AA from buccal tissue: 0.085 (95 % CI 0.014 to 0.156; p=0.02) years of AA per 1 kg/m$^2$ increase in BMI. The strength of association was lower for blood tissue: 0.044 years of AA per 1 kg/m$^2$ change in BMI (95 % CI −0.065 to 0.154 years; p=0.42) (table 3). There was no association between height and AA. We observed an association between smoking and AA of buccal tissue (p=0.001), but not in blood tissue. Current smokers had the lowest AA on average, with ex-smokers and never-smokers having 1.88 (95 % CI 0.85 to 2.9) and 1.85 (95 % CI 0.76 to 3.0) extra years of AA, respectively. AA did not vary by childhood or adult SEP.

### Sensitivity analysis
We provide associations of AA and change in physical capability when including imputed data for those unable to perform tests in online supplementary table S3. Including individuals unable to perform the grip strength test attenuates its association with AA. For a 1 year increase in AA, grip strength decreased by an additional 0.29 kg (95 % CI −0.74 to 0.15 kg; p=0.19) from age 53 to 60–64 in a fully adjusted model. Including individuals unable to perform tasks did not dramatically affect any of the other blood AA or buccal associations.

### Replication
Using data from 988 ALSPAC women with mean age 46.9, we attempted to replicate the cross-sectional findings from NSHD (tables 4 and 5). The median absolute error and correlation between epigenetic and chronological age was 3.9 years and 0.53 in ALSPAC, respectively, where the chronological age was 47.4 (SD 4.5 years, range 34.5–60). AA was not related to grip strength, chair rise speed or balance time. The finding that higher BMI was associated with greater AA was replicated in the ALSPAC women (0.129 years per 1 kg/m$^2$ increase in BMI, 95 % CI 0.051 to 0.207 years, p=0.001) but smoking was not associated with AA (p=0.43), although the direction of effect

**Table 2**  Association of age acceleration with changes in physical capability from age 53 to 64 in NSHD participants

| Variable | Model* | Blood (n=152) | | | Buccal (n=790) | | |
|---|---|---|---|---|---|---|---|
| | | Regression coefficient (difference per year AA) | 95% CI | p Value | Regression coefficient (difference per year AA) | 95% CI | p Value |
| Grip strength (kg) | Unadjusted | −0.34 | −0.70 to 0.01 | 0.06 | −0.02 | −0.16 to 0.12 | 0.73 |
| | Adjusted for age, height, BMI | −0.33 | −0.69 to 0.02 | 0.06 | −0.03 | −0.17 to 0.12 | 0.73 |
| | Adjusted for height, BMI, smoking, education and SEP | −0.42 | −0.82, to 0.03 | 0.03 | −0.07 | −0.22 to 0.08 | 0.35 |
| Chair rise speed (stands/min) | Unadjusted | 0.20 | −0.23 to 0.62 | 0.37 | −0.03 | −0.17 to 0.12 | 0.70 |
| | Adjusted for age, height, BMI | 0.19 | −0.24 to 0.62 | 0.38 | −0.04 | −0.19 to 0.10 | 0.58 |
| | Adjusted for height, BMI, smoking, education and SEP | 0.06 | −0.40 to 0.52 | 0.80 | −0.05 | −0.20 to 0.10 | 0.53 |
| Balance time, eyes closed (log-seconds) | Unadjusted | −0.01 | −0.04 to 0.02 | 0.38 | −0.001 | −0.011 to 0.010 | 0.92 |
| | Adjusted for age, height, BMI | −0.01 | −0.04 to 0.02 | 0.39 | −0.002 | −0.012 to 0.009 | 0.76 |
| | Adjusted for height, BMI, smoking, education and SEP | −0.01 | −0.04 to 0.02 | 0.55 | −0.002 | −0.012 to 0.009 | 0.73 |
| Composite score | Unadjusted | −0.01 | −0.028 to 0.003 | 0.10 | 0.001 | −0.005 to 0.007 | 0.77 |
| | Adjusted for age, height, BMI | −0.01 | −0.028 to 0.003 | 0.11 | 0.000 | −0.005 to 0.006 | 0.87 |
| | Adjusted for height, BMI, smoking, education and SEP | −0.01 | −0.03 to 0.01 | 0.16 | −0.001 | −0.007 to 0.005 | 0.68 |

AA, age acceleration; BMI, body mass index; SEP, socioeconomic position.
*For each of the four physical capability outcome measures, we ran three models, first unadjusted, then adjusted for height and BMI, then adjusted for height, BMI, smoking, education and both adult and childhood SEP.

was the same, with ex-smokers having 0.56 years higher AA and never smokers 0.17 years higher AA compared with current smokers on average. As in NSHD, height and education were not associated with AA.

## DISCUSSION

Age acceleration in blood is associated with a greater decline in grip strength from age 53 to 60–64; for every 1 year of AA, women had a 0.4 kg greater decrease in grip strength. Neither blood nor buccal epigenetic age acceleration at age 53 was associated with grip strength or other measures of physical capability at either age 53 or 60–64. The epigenetic age calculated in our sample was systematically lower than chronological age, particularly for buccal cells (mean difference of 10.7 years for buccal cells).

Our study is one of the first to examine epigenetic age from different tissues on the same individuals in relation to risk factors for mortality. We used serial measures of physical capability on the same individuals over time, allowing for better inferences on changes in physical capability in late midlife, compared with having just cross-sectional data. However, one limitation is having just two measures, which are susceptible to regression to the mean. A limitation of our findings is the lack of generalisability—the subsample of the cohort consisted of females only with repeated measures at ages 53 and 64 and was restricted to those with complete information on particular variables of interest (ie, blood and buccal samples). The 790 women sampled here had marginally lower grip strength (25.7 vs 26.0 kg, p=0.4) and chair rise speed (24.5 vs 25.5 stands/min, p=0.007) than NSHD

**Table 3** Associations of mortality risk factors with outcome of age acceleration (years) at 53 for NSHD participants*

| Variable | Level | Blood (n=152) | | | Buccal (n=790) | | |
|---|---|---|---|---|---|---|---|
| | | Regression coefficient (difference per year AA) | 95% CI | p Value | Regression coefficient (difference per year AA) | 95% CI | p Value |
| Height (cm) | | −0.017 | −0.085 to 0.051 | 0.63 | −0.011 | −0.14 to 0.11 | 0.86 |
| BMI 53 (kg/m$^2$) | | 0.085 | 0.014 to 0.16 | 0.02 | 0.044 | −0.065 to 0.15 | 0.42 |
| Smoking | Current | Reference | | 0.001 | Reference | | 0.42 |
| | Ex-smoker | 1.88 | 0.85 to 2.91 | | 0.83 | −0.99 to 2.66 | |
| | Never | 1.86 | 0.76 to 2.95 | | −0.16 | −2.17 to 1.85 | |
| Childhood SEP | Professional | Reference | | 0.80 | Reference | | 0.56 |
| | Intermediate | −0.73 | −2.44 to 0.99 | | −1.30 | −4.01 to 1.40 | |
| | Skilled (non-manual) | −0.81 | −2.59 to 0.97 | | −0.48 | −3.24 to 2.27 | |
| | Skilled (manual) | −0.25 | −1.89 to 1.38 | | −1.24 | −3.88 to 1.39 | |
| | Partly skilled | −0.94 | −2.67 to 0.79 | | −0.11 | −2.90 to 2.67 | |
| | Unskilled | −0.40 | −2.57 to 1.78 | | 1.39 | −2.40 to 5.17 | |
| Adult SEP | Professional | Reference | | 0.62 | Reference | | 0.35 |
| | Intermediate | 0.95 | −2.11 to 4.01 | | −1.65 | −10.27 to 6.97 | |
| | Skilled (non-manual) | 0.95 | −2.10 to 4.00 | | −1.95 | −10.59 to 6.69 | |
| | Skilled (manual) | 1.35 | −1.97 to 4.68 | | −0.25 | −9.13 to 8.63 | |
| | Partly skilled | 0.032 | −3.12 to 3.18 | | −3.41 | −12.31 to 5.50 | |
| | Unskilled | 0.52 | −2.85 to 3.89 | | 0.24 | −8.73 to 9.22 | |
| Education | None | Reference | | 0.78 | Reference | | 0.64 |
| | Vocational | 0.16 | −1.65 to 1.98 | | −1.25 | −4.72 to 2.22 | |
| | Sub-GCSE | 1.20 | −0.73 to 3.14 | | −1.74 | −5.21 to 1.73 | |
| | O level | 0.41 | −0.63 to 1.45 | | −1.02 | −2.90 to 0.86 | |
| | A level | −0.53 | −1.81 to 0.75 | | −1.97 | −4.29 to 0.35 | |
| | Burham A2 | 0.55 | −0.91 to 2.00 | | −1.43 | −3.66 to 0.81 | |
| | Degree | −0.28 | −2.36 to 1.81 | | −2.01 | −5.73 to 1.71 | |
| | Postgraduate | −0.81 | −8.73 to 7.11 | | 0.84 | −0.38 to 2.06 | |

*AA, age acceleration; BMI, body mass index; GCSE, General Certificate of Secondary Education; NSHD, National Survey of Health and Development; SEP, socioeconomic position.

women overall at age 60–64.[36] Including those individuals who were unable to perform the grip strength test attenuated the association with blood AA. Although 18 and 21 grip strength tests could not be performed at age 53 and 60–64, respectively (table 1), just 6 of these were from individuals with blood DNA methylation available. The attenuation was mainly due to a single individual with low AA and high grip strength at age 53 who was unable to perform the test at age 60–64. Our results should be viewed with consideration for multiple testing. Our primary analysis includes epigenetic age acceleration from two tissues tested against four measures of physical capability, giving a total of eight tests. This diminishes the strength of the evidence provided by this study.

Our blood results should be compared with another recent study of epigenetic age and physical capability in an older UK birth cohort.[17] Using data from the Lothian Birth Cohort 1936, cross-sectional associations at age 70 were found between greater epigenetic age and weaker grip strength as well as with lower lung function and cognitive capability, but not walking speed; however, they found no association between baseline epigenetic age and changes in either physical or cognitive capability from age 70 to 76. We have found some evidence that epigenetic age measured at 53 may be associated with a greater decline in grip strength between age 53 and 60–64, but no associations were identified with any physical capability measure at age 53 or 60–64. In the

**Table 4** Replication of physical capability and age acceleration in ALSPAC mothers with mean age 46.9

| Variable | Model* | Regression coefficient (difference per year AA) | 95% CI | p Value |
|---|---|---|---|---|
| Grip strength (kg) | Unadjusted | −0.021 | −0.12 to 0.078 | 0.68 |
| | Adjusted for height, BMI | −0.044 | −0.14 to 0.052 | 0.37 |
| | Adjusted for height, BMI, SEP | −0.034 | −0.13 to 0.065 | 0.50 |
| Chair rise speed (stands/min) | Unadjusted | 0.0004 | −0.0004 to 0.001 | 0.31 |
| | Adjusted for height, BMI | 0.0005 | −0.0003 to 0.001 | 0.22 |
| | Adjusted for height, BMI, SEP | 0.0004 | −0.0004 to 0.001 | 0.31 |
| Balance time, eyes closed (log-seconds) | Unadjusted | 0.067 | −0.058 to 0.19 | 0.30 |
| | Adjusted for height, BMI | 0.083 | −0.045 to 0.21 | 0.20 |
| | Adjusted for height, BMI, SEP | 0.088 | −0.042 to 0.22 | 0.18 |

AA, age acceleration; ALSPAC, Avon Longitudinal Study of Parents and Children; BMI, body mass index; SEP, socioeconomic position.
*For each of the four physical capability measures, we ran three models, first unadjusted, then adjusted for height and BMI, then adjusted for height, BMI, smoking and education.

Lothian Birth Cohort, the reported effect size of AA on grip strength at age 70 was −0.05 kg per year of blood AA with a sample size of 100.[17] In our analysis, the effect sizes at age 53 and 60–64 were 0.18 and −0.15 kg per year of blood AA, respectively with a sample size of 152. The relatively small sample in our analysis of blood AA may mean our study lacks the required power, or it could be that this association only manifests at older ages. It is possible that the association is beginning to emerge in NSHD at age 60–64 (an age still younger than the Lothian Birth Cohort baseline), with the suggestion of faster rates of change in those with greater AA. We do not, however, observe any associations between buccal cell AA and physical capability where we have a larger sample size and more comparable statistical power to the Lothian Birth Cohort. Our different results could also be due to sex differences between the two studies; the LBC includes both men and women, whereas we have only looked in females. Several studies have identified higher AA in men than in women.[37] To better understand the

epigenetic embodiment of physical capability in later life, one might perform epigenome-wide analysis of these measures.

The weak correlations found between epigenetic and chronological ages (0.022 for blood, 0.115 for buccal cells) should be considered with the knowledge that the SD of age is 0.16 years (range 53–54 years). Horvath,[1] using data from across 82 studies, compared the SD of age measured in each study with the correlation coefficient found in each study between epigenetic and chronological age. He found a correlation of 0.49 between the SD of age and the performance of his epigenetic clock (in terms of correlation). Thus, with such a small age range in our sample, it should be no surprise that we find a diminished correlation. In ALSPAC, by comparison, where the SD of age is larger at 4.5 years, the correlation between epigenetic and chronological age is 0.53. Comparing the median absolute error the difference is much smaller, with 4.1 years in NSHD and 3.9 years in ALSPAC. This suggests that while the correlation metric is not suitable

**Table 5** Replication of mortality risk factors with outcome of age acceleration in ALSPAC mothers with mean age 46.9

| Variable | Level | Association with AA (years) | 95% CI | p Value |
|---|---|---|---|---|
| Height (cm) | | 0.0288 | −0.034 to 0.092 | 0.37 |
| BMI (kg/m$^2$) | | 0.1293 | 0.051 to 0.21 | 0.001 |
| Smoking | Current | Reference | | 0.43 |
| | Ex | 0.5605 | −0.88 to 2.00 | |
| | Never | 0.1693 | −1.90 to 2.24 | |
| Education | Secondary | Reference | | 0.28 |
| | Vocational | 0.1693 | −1.90 to 2.24 | |
| | O level | 0.2586 | −1.34 to 1.86 | |
| | A level | −0.0295 | −1.65 to 1.59 | |
| | Degree | 1.2266 | −0.47 to 2.93 | |

AA, age acceleration; ALSPAC, Avon Longitudinal Study of Parents and Children; BMI, body mass index.

for NSHD, the epigenetic clock itself is valid for blood samples.

While the epigenetic clock was trained on observations from individuals from newborn to 100 years of age, it is likely that age-specific clocks could improve on Horvath's clock. However, these benefits are negated by the loss of generalisability. It is also required that the relationship between chronological age and epigenetic age is linear. In our sample, there is no evidence against a linear relationship, and the residuals from this model of epigenetic age on chronological age, that is, the age accelerations themselves, were normally distributed in this older population.

While the very small range of age at which the epigenetic information was taken in our sample explains the low correlation, it does not explain the bias when using the buccal samples. The systematic difference found here appears to be related to tissue specificity, since epigenetic age from blood was closer to chronological age than buccal epigenetic age. The Horvath age estimator was developed using publicly available data covering 51 tissue types (including buccal cells) such that tissue specificity should not result in such an underestimate of chronological age. Three sets of publicly available buccal cell DNA methylation data[38–40] were among those used in the development of the Horvath epigenetic clock, with a reported correlation of 0.9 between chronological and buccal epigenetic age.[1] However, these were from 109 adolescents,[38] 30 newborns (ie, 10 pairs of monozygotic (MZ) and 5 pairs of dizygotic (DZ) twins between birth and 18 months[40]) and 10 individuals who were aged 16, 27, 28, 29, 37, 42, 44, 44, 52, 68[39]. The systematic difference (of 10.7 years) may be explained by the lack of overlap in the age at which information from buccal cells was available in our study and those in the training dataset used to derive the epigenetic clock. Our study questions the use of the epigenetic clock for buccal samples in females aged between 53 and 60–64.

There are a growing number of studies comparing DNA methylation from more than one tissue on the same group of individuals,[10 25 41] with one of these using the same NSHD data as the current study.[10] One novel application of the epigenetic clock is to estimate epigenetic ages from different tissues on the same individuals. In the current study, we have found evidence that buccal samples are epigenetically younger than blood samples in a UK population. We found a weak correlation between epigenetic age from blood and buccal cells in the same individuals (r=0.19). This low correlation may be due to residual confounding. Since the blood samples come from a case-control study within the birth cohort from which the 790 buccal samples were taken, it may be that the poor correlation between tissues is attributable to selection bias. Further comparisons of epigenetic ages from different tissue types in the same individuals may elucidate our findings.

We found both lower BMI and smoking were related to age deceleration, with the BMI finding replicated in blood methylation age from ALSPAC women at mean age

46.9. Previous research has found this same association between higher BMI and AA in liver tissue,[7 37] but ours is the first finding in buccal cells. Our finding that smoking is associated with lower AA is unexpected, since previous research suggests that positive blood AA is associated with higher rates of mortality.[11] This could be due to our use of buccal cells, which are likely more reflective of the effect of smoking on DNA methylation.[25] It is currently unknown whether buccal AA is associated with mortality, nor if the direction is the same as blood AA. Further to this point, the smoking and AA association was not seen in blood samples from ALSPAC or NSHD participants, suggesting this particular result may be spurious.

In conclusion, having a higher epigenetic than chronological age is associated with a greater decline in grip strength in British females aged between 53 and 60–64, but overall there is little evidence that AA is associated with physical capability change in these women. AA does not appear to be related to measures of physical capability in women at ages 53 or 60–64, while BMI appears to be associated with accelerated epigenetic age in this population.

**Author affiliations**
[1]MRC Integrative Epidemiology Unit at the University of Bristol, Population Health Sciences, Bristol Medical School, University of Bristol, Bristol, UK
[2]MRC Unit for Lifelong Health and Ageing at UCL, London, UK
[3]Department of Women's Cancer, University College London, London, UK
[4]CAS Key Lab of Computational Biology, CAS-MPG Partner Institute for Computational Biology, Shanghai Institute for Biological Sciences, Chinese Academy of Sciences, Shanghai, China
[5]Department of Statistical Cancer Genomics, UCL Cancer Institute, University College London, London, UK
[11]Insight Centre for Data Analytics, National University of Ireland, Galway, Galway, Ireland

**Acknowledgements** For use of the ALSPAC data, the authors are extremely grateful to all the families who took part in this study, the midwives for their help in recruiting them and the whole ALSPAC team, which includes interviewers, computer and laboratory technicians, clerical workers, research scientists, volunteers, managers, receptionists and nurses.

**Contributors** This publication is the work of the authors and AJS will serve as guarantors for the contents of this paper. AJS performed, designed and ran analysis, drafted the manuscript. RC, RH, DK, AW, AT, MW supervised NSHD analysis. GDS, CLR, LDH supervised ALSPAC analysis. All authors contributed to the final draft manuscript.

**Funding** This work was supported by a grant from the UK Economic and Social Research Council (ES/M010317/1) and by the UK Medical Research Council (programme codes MC_UU_12019/1, MC_UU_12019/2, MC_UU_12019/4). Research reported in this publication was supported by the National Institute on Ageing of the National Institutes of Health under Award No. R01AG048835. AJS was supported by the UK Medical Research Council (grant MR/L011824/1). LDH was supported by a fellowship from the UK Medical Research Council (MR/M020894/1). The UK Medical Research Council and the Wellcome Trust (grant ref: 102215/2/13/2) and the University of Bristol provide core support for ALSPAC. Collection of ALSPAC epigenetic and phenotypic data were funded by grants from the Wellcome Trust (WT092830M) and Joint UK Research Councils under th Lifelong Health and Wellbeing—Phase III initiative (G1001357).

**Competing interests** None declared.

**Ethics approval** Ethical approval for ALSPAC was obtained from the ALSPAC Ethics and Law Committee and the Local Research Ethics Committees. The content is solely the responsibility of the authors and does not necessarily represent the official views of the National Institutes of Health.

**Provenance and peer review** Not commissioned; externally peer reviewed.

**Data sharing statement** Data used in this publication are available to bona fide researchers on request to the NSHD Data Sharing Committee via a standard application procedure. Further details can be found at: http://www.nshd.mrc.ac.uk/data; doi: 10.5522/NSHD/Q101; doi: 10.5522/NSHD/Q102.

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
