## [Reviewer comments · BMJ Open]

ARTICLE DETAILS

TITLE (PROVISIONAL)	Are objective measures of physical capability related to accelerated epigenetic age? Findings from a British birth cohort.
AUTHORS	Simpkin, Andrew; Cooper, Rachel; Howe, Laura; Relton, Caroline; Davey Smith, George; Teschendorff, Andrew; Widschwendter, Martin; Wong, Andrew; Kuh, Diana; Hardy, Rebecca

VERSION 1 - REVIEW

REVIEWER	Christian Page Oslo Centre for Biostatistics and Epidemiology Division for Research Support Oslo University Hospital
REVIEW RETURNED	31-Mar-2017

GENERAL COMMENTS	Review of Simpkin et.al. This manuscript looks into the epigenetic clock, more specifically the age acceleration and compare it to physical measurement of aging (grip strength, chair rise speed). This adds to the research on the biology of aging and the differences between biological age and chronological age. The authors also acknowledge and discuss the limitations of the study well, including the fact that only one individual were over 53 years in the linear predictor from Horvat, and the very narrow age range of the participants, which probably explains the observed weak correlations between age and the epigenetic clock. With some minor adjustments to the manuscript, I consider it acceptable for publication. Minor revisions: i. I think that the authors should include a broader discussion on the validity of the age prediction in an older population using a linear predictor. It is likely that the difference between the epigenetic (biological clock) and the chronological age diverges with age above a certain age. And that the assumption of linearity or that other probes or genes is a better predictor of biological age.ii. The authors provide a careful interpretation of the results and don't overstate the significant p-values. They do not, however, provide p-values adjusted for multiple testing, and I'm missing a good rationale for not giving the adjusted p-values.iii. The authors should also consider to include the correlation between the epigenetic age and chronological age in the ASLPAC moms, which I think would be higher for the NSHD cohort, due to the wider age range.iv. Numerus brackets are not matched in the manuscript (e.g. page 2 line 43; page 3 line 26; page 10 line 50).
---

REVIEWER	Albert Merrill Levin Henry Ford Health System United States of America
REVIEW RETURNED	17-Apr-2017

GENERAL COMMENTS	The authors present an interesting manuscript on findings from a study of a methylation-based predictor of age and physical capability changes over time. While the study has multiple strengths (prospective cohort study, DNA methylation age from multiple tissues, and a replication study), there are open questions about the actual accuracy and validity of the methylation-based age model in this study that could not be fully evaluated given the data presented. These points coupled with an under-described sampling scheme(s) made it difficult to evaluate the full merits of the manuscript. These may be able to be addressed in revision. My principal comments on these points and others are presented below with the intent to aid in the process of revision.  1. On page 6, line 11 of the Methods, the authors introduce the 790 female subjects that the study was based on. However, there are no details given on how these women were sampled. Was this a random sample from the NSHD? It seems that this might not be the case as approximately half of those with both blood and buccal swab DNA had a cancer diagnosis after the age of 53. Regardless, more details need to be given in order for the reader to fully understand what group of people these results are applicable to (i.e. the population of inference). 2. There are no details given regarding the processing of the Illumina 450k DNA methylation chips, quality control standards applied, and normalization procedures. This information is critical as the methylation-based epigenetic age model of Horvath seems to be downwardly biased in this population. Could this somehow be due to the processing of the methylation data upstream of the estimation of methylation-based age? The reader can't evaluate this if the actual processing steps are not given. 3. This leads to the most critical point, which is the accuracy and validity of the epigenetic age predictor in this study. The authors need to provide a more appropriate evaluation of the fit of the predicted age to actual age in this study. As Horvath pointed out in his manuscript describing the development of this age prediction model, the Pearson correlation is not an appropriate measure to assess the calibration of the model when all of the participants are essentially the same age, 53 years in this case. The authors should present the median absolute deviation of methylation predicted age vs. actual age as a measure of accuracy. However, given that the average predicted ages are significantly lower than the mean actual age for both buccal and blood predictors, this suggests that the error will be substantial, especially for the buccal measures. This perceived error calls into question the applicability (i.e. validity) of the Horvath model for this particular data set and therefore all of the results that the methylation age is based upon. The authors need to give a more in-depth assessment of the accuracy of the Horvath method prediction in this dataset as well as a better characterization of why the predictions seem better for the methylation data from blood in comparison to the buccal DNA.
--

	4. It is not clear from the Methods exactly how the age acceleration in the blood was adjusted based on this Houseman method. Was a model of the age acceleration residuals constructed, where the Houseman cell composition estimates were adjusted for producing the actual age acceleration residuals, or were the cell composition estimates included as covariates in the models of physical capacity along with the other covariates for blood-based age acceleration models? 5. Given that the Horvath method is supposed to be a predictor of age regardless of cell type, it's interesting that the results are so different between buccal and blood based DNA measures of age acceleration. One possible reason could be residual confounding. Both blood and buccal samples are treated essentially the same in the analysis, but half of the blood sample are from individuals who subsequently developed cancer, indicating a case-control sampling for this sample subset. If this sampling scheme is not accounted for in the analysis, the results might be due to this selection and not age acceleration. Again, the lack of a complete description of the sampling scheme(s) (point #1) makes it difficult for the reader to assess the appropriateness of the methods applied and the accuracy of the results. 6. Also, the replication population description is missing key points. In particular, how were the mothers selected, how was methylation assessed, and from what cells was the DNA assessed (buccal, blood, or some other tissue)? The reader should not have to read other articles to determine these critical pieces of information. 7. On page 8, line 42, what is meant by the term "minimum sex-specific value"? The study was only conducted in females. So, are the authors using the minimum value in the study or some other reference to determine the minimum value in women? 8. On page 10, line 50, there is an incomplete sentence. 9. In general, the abbreviations are left undefined in the Tables, and some are left undefined in the text (eg. "MRC").
--	--

REVIEWER	Dr. Faisal I Rezwan University of Southampton
REVIEW RETURNED	27-Apr-2017

GENERAL COMMENTS	This paper focuses on the study to find the association between age acceleration (AA, the difference between chronological and epigenetic age) and physical capability measures in mainly female subjects aged over 50 years. It also presented an extension by exploring the association between AA and some life measures. Here authors have measured epigenetic age from buccal and blood samples, which is an interesting approach to understand the variation in epigenetic age measures due to different tissue types though Horvath method is independent of tissue-specificities. Authors have identified association between AA and grip strength. However, the direction of association reversed with ~10 years difference. The paper is well written and easy to read (except few places, mentioned below), and existing literature is well explored and cited. However, I have some major concerns that I feel are
---

required to be addressed and this will help the authors to improve the article.

Major comments:

1. I have found the title of the paper quite obscure as it lacks in the reflection of the whole research. The title is very broad and does not represent the study undertaken and explained in the paper.

2. It is understandable that anyone would find the study very much sex-specific as only female subjects were used and authors have rightly addressed this limitation. However, while presenting the result the authors put strong decision based on inconclusive results. For example: Conclusion in Abstract section stated that evidence have been found on AA associated with greater decrease of grip strength. However, this result is for female only. Therefore, though the use female subject only was mentioned in the limitation, in conclusion that must have been emphasised.

3. In the objective, authors mentioned about investigating the association between epigenetic age and physical capability in later life. However, actually investigation on the association between AA, not epigenetic age, and objective measures of physical capabilities was performed. Therefore, authors really need clarify their aims/objectives properly.

4. For socio economic position, it is not clear why authors used paternal occupational class. A normal trend is always to use maternal socio-economic status or educational level. In Method section, it was also not clear whether the socio-economic position was the combination of paternal and subjects' socio-economic positions or not. However, this has been cleared in the statistical analysis section. I would recommend to clarify this in the Method section.

5. Authors has mentioned height, BMI, smoking and SEP as mortality risk factors in the Statistical Analysis section. However, it was not explained how they fixed these as mortality risk factors. A reference in this case is required.

6. In the last paragraph of the Statistical Analysis section (Page 8, Lines 37 -42), authors mentioned about using minimum sex-specific values for imputation. It is not clear what does this mean. I would recommend to use some imputation methods (such as: MICE) for imputation, not just replacing missing values with minimum values.

7. The correlation between chronological age and epigenetic ages (both from blood and buccal) is very low. Horvath's online clock generally generates error or warning messages, which sometimes worth to check. It looks like the 353 CpG sites, required for calculating epigenetic age, may have low quality methylation values in those samples. While authors put a reasonable discussion why this may happen in case of buccal tissue, epigenetic age from blood samples shows that the age of samples does not really matter if they are adult. The correlation coefficient between epigenetic age and chronological age is still very low and this is alarming.

8. It was not clear whether authors checked for outliers and discarded them or not. For mentioned sample sizes, some outliers may really dominate the regression outcome. Therefore, I would recommend the authors to check for outliers, discard them and re-run the analyses.

9. As mentioned above, authors have made some strong comments on the finding of the study based on inconclusive results. The results between different age groups are inconsistent and replication cohort could not show the same findings. For example: in Conclusion, authors mentioned that AA is associated with a greater decline of grip strength in middle age and the term "middle age" has been used

	loosely here. Minor comments: 1. Page 3 Point from Line 21 is incomplete. 2. First paragraph of Discussion Page 13 Line 6 -17, is not clearly written. I would recommend rewriting/rephrasing that part specifically from Lines 8 -12. 3. Page 14 Line 16, unit of grip strength “kg” is missing.
--	---

VERSION 1 – AUTHOR RESPONSE

Reviewer: 1

Reviewer Name: Christian Page

Institution and Country: Oslo Centre for Biostatistics and Epidemiology, Division for Research Support, Oslo University Hospital

Please state any competing interests: None declared

Please leave your comments for the authors below

Review of Simpkin et.al.

This manuscript looks into the epigenetic clock, more specifically the age acceleration and compare it to physical measurement of aging (grip strength, chair rise speed). This adds to the research on the biology of aging and the differences between biological age and chronological age.

The authors also acknowledge and discuss the limitations of the study well, including the fact that only one individual were over 53 years in the linear predictor from Horvat, and the very narrow age range of the participants, which probably explains the observed weak correlations between age and the epigenetic clock. With some minor adjustments to the manuscript, I consider it acceptable for publication.

Minor revisions:

i. I think that the authors should include a broader discussion on the validity of the age prediction in an older population using a linear predictor. It is likely that the difference between the epigenetic (biological clock) and the chronological age diverges with age above a certain age. And that the assumption of linearity or that other probes or genes is a better predictor of biological age.

Thank you for this comment. The linearity assumption through the elastic net was tested across the 82 cohorts in the original Horvath paper, but indeed this may not be true in all cases. We have added further discussion of the epigenetic clock in this older population:

“While the epigenetic clock was trained on observations from individuals from new-born to 100 years of age, it is likely that age specific clocks could improve on Horvath’s clock. However, these benefits are negated by the loss of generalisability. It is also required that the relationship between chronological age and epigenetic age is linear. In our sample, there is no evidence against a linear relationship in Figure 1, and the residuals from this model of epigenetic age on chronological age, i.e. the age accelerations themselves, were normally distributed in this older population.”

ii. The authors provide a careful interpretation of the results and don’t overstate the significant p-values. The do not, however, provide p-values adjusted for multiple testing, and I’m missing a good rationale for not giving the adjusted p-values.

Thank you for this comment and agree that multiple testing is a limitation of our manuscript. We have run eight tests in our primary analysis so a threshold for statistical significance should be $0.05/8 = 0.00625$. However, since we avoid the term statistical significance in our manuscript, the use of a Bonferroni correction for significance does not fit with the strength of evidence paradigm employed here. To make the reader more aware we have added a discussion point on the limitation of our evidence in view of the number of tests:

“Our results should be viewed with consideration for multiple testing. Our primary analysis includes epigenetic age acceleration from two tissues tested against four measures of physical capability, giving a total of eight tests. This diminishes the strength of the evidence provided by this study.”

iii. The authors should also consider to include the correlation between the epigenetic age and chronological age in the ALSPAC moms, which I think would be higher for the NSHD cohort, due to the wider age range.

We have added the median absolute error (3.9 years) and correlation between epigenetic and actual age (0.53) in the results section and then used this to add to our discussion of poor performance in the NSHD cohort. We thank the reviewer for this point which has very much assisted the view of correlation in this manuscript.

Results:

“The median absolute error and correlation between epigenetic and chronological age was 3.9 years and 0.53 in ALSPAC respectively, where the chronological age was 47.4 (standard deviation 4.5 years, range 34.5 to 60).”

Discussion:

“The weak correlations found between epigenetic and chronological ages (0.022 for blood, 0.115 for buccal cells) should be considered with the knowledge that the standard deviation of age is 0.16 years (range 53-54 years). Horvath¹, using data from across 82 studies, compared the standard deviation of age measured in each study with the correlation coefficient found in each study between epigenetic and chronological age. He found a correlation of 0.49 between the SD of age and the performance of his epigenetic clock (in terms of correlation). Thus, with such a small age range in our sample, it should be no surprise that we find a diminished correlation. In ALSPAC, by comparison, where the SD of age is larger at 4.5 years, the correlation between epigenetic and chronological age is 0.53. Comparing the median absolute error the difference is much smaller, with 4.1 years in NSHD and 3.9 years in ALSPAC. This suggests that while the correlation metric is not suitable for NSHD, the epigenetic clock itself is valid for blood samples.”

iv. Numerus brackets are not matched in the manuscript (e.g. page 2 line 43; page 3 line 26; page 10 line 50).

Thank you, we have fixed the parentheses.

Reviewer: 2

Reviewer Name: Albert Merrill Levin

Institution and Country: Henry Ford Health System, United States of America

Please state any competing interests: None declared

Please leave your comments for the authors below

The authors present an interesting manuscript on findings from a study of a methylation-based predictor of age and physical capability changes over time. While the study has multiple strength

(prospective cohort study, DNA methylation age from multiple tissues, and a replication study), there are open questions about the actual accuracy and validity of the methylation-based age model in this study that could not be fully evaluate given the data presented. These points coupled with an under-described sampling scheme(s) made it difficult to evaluate the full merits of the manuscript. These may be able to be addressed in revision. My principal comments on these points and others are presented below with the intent to aid in the process of revision.

1. On page 6, line 11 of the Methods, the authors introduce the 790 female subjects that the study was based on. However, there are no details given on how these women were sampled. Was this a random sample from the NSHD? It seem that this might not be the case as approximately have of those with both blood and buccal swab DNA had a cancer diagnosis after the age of 53. Regardless, more details need to be given in order to for the reader to fully understand what group of people these results are applicable to (i.e. the population of inference).

Thank you for this comment. Our study includes all DNA methylation data available in the NSHD study but we see that the description of our sample was lacking. The 790 women included here had biological samples collected in 1999 when DNA methylation data were not the primary goal of collection. Indeed, those with blood available were sampled as part of a case-control study of cancer. We have clarified this in our Study participants section:

“DNA was first collected in NSHD participants at age 53 in 1999; following QC, the study sample with epigenetic information available consisted of 790 women who had a buccal cell sample taken at age 53 (mean age 53.4, standard deviation 0.16, range 53 to 54) , and who had complete information on epidemiological variables of interest and follow-up. Among these 790 women there were 152 who also had epigenetic information available from blood at age 53. These 152 were originally selected for a case-control study of cancer, and consisted of 75 incident cancer cases after age 53 and 77 controls randomly selected from those with complete data available for the cancer study 16 25 26”

“DNA methylation was measured using the Infinium HumanMethylation450 BeadChip (Illumina, Inc) in NSHD participants who had biological samples collected in 1999; 638 (buccal cell only) and 152 (buccal cell and blood)6.”

2. There are no details given regarding the processing of the Illumina 450k DNA methylation chips, quality control standards applied, and normalization procedures. This information is critical as the methylation-based epigenetic age model of Horvath seems to be downwardly biased in this population. Could this somehow be due to the processing of the methylation data upstream of the estimation of methylation-based age? The reader can't evaluate this if the actual processing steps are not given.

We have now included a detailed description of the processing of our data:

“DNA methylation was measured using the Infinium HumanMethylation450 BeadChip (Illumina, Inc) in NSHD participants who had biological samples collected in 1999; 638 (buccal cell only) and 152 (buccal cell and blood) 6. Quality control and normalisation was performed on each of the 790 buccal and then separately on the 152 matched whole blood samples. For each, the minfi package was used to process raw .idat data files⁸, using the Illumina definition of beta-values and extracting P-values of detection for each sample. The Illumina methylation beta-value of a given CpG site is found from the intensity of the methylated (M) and unmethylated (U) alleles, as the ratio of fluorescent signals $\beta = \text{Max}(M,0) / [\text{Max}(M,0) + \text{Max}(U,0) + 100]$. The level of methylation is expressed as a “beta” value (β -value), ranging from 0 (no cytosine methylation) to 1 (complete cytosine methylation). As a further QC step, probes that contained <95% of signals detectable above background signal (detection p-value<0.01) were removed from further analysis, and the rest of missing values were imputed using

the k-nearest neighbours imputation procedure⁹. To correct for the well-known bias of type-2 probes, we used the SWAN package¹⁰. To check robustness of this correction procedure, we verified that results were largely unchanged using BMIQ¹¹. This completed the intra-sample normalization. All participants provided written informed consent. The Central Manchester Ethics Committee approved the use of these samples for epigenetic studies of health.”

3. This leads to the most critical point, which is the accuracy and validity of the epigenetic age predictor in this study. The authors need to provide a more appropriate evaluation of the fit of the predicted age to actual age in this study. As Horvath pointed out in his manuscript describing the development of this age prediction model, the Pearson correlation is not an appropriate measure to assess the calibration of the model when all of the participants are essentially the same age, 53 years in this case. The authors should present the median absolute deviation of methylation predicted age vs. actual age as a measure of accuracy. However, given that the average predicted ages are significantly lower than the mean actual age for both buccal and blood predictors, this suggests that the error will be substantial, especially for the buccal measures. This perceived error calls into question the applicability (i.e. validity) of the Horvath model for this particular data set and therefore all of the results that the methylation age is based upon. The authors need to give a more in-depth assessment of the accuracy of the Horvath method prediction in this dataset as well as a better characterization of why the predictions seem better for the methylation data from blood in comparison to the buccal DNA.

We have now included the median error as our main measure of accuracy. By using the same measure in ALSPAC we have shown that the epigenetic clock appears valid in the blood samples from both NSHD and ALSPAC, but that as the reviewer rightly points out the validity is questionable for the buccal samples. We have added this to our discussion as well as highlighting median error as the main measure in our results:

Methods:

“Median absolute error was used to investigate the relationship between chronological age and epigenetic age from blood and buccal tissue, with correlation being secondary given the low range of actual age in NSHD.”

Results:

“The descriptive statistics are displayed in Table 1, with epigenetic ages shown in Figures 1. The average epigenetic age was 42.8 (SD 5.71 years) using DNA methylation from buccal tissue and 50.3 (SD 4.34 years) using DNA methylation from blood. These both underestimate the average chronological age of 53.4 years (SD 0.16 years). The median absolute error between chronological and epigenetic age is 10.5 and 4.1 years for buccal and blood respectively. Correlation with chronological age was much lower than previously reported: 0.022 ($p=0.79$) for blood age and 0.115 ($p=0.16$) for buccal age, but this correlation is a less appropriate assessment due to the narrow age range (SD of age=0.16 years). Correlation was slightly higher between the two epigenetic ages, with a Pearson correlation coefficient of 0.190 ($p=0.02$).”

“The median absolute error and correlation between epigenetic and chronological age was 3.9 years and 0.53 in ALSPAC respectively, where the chronological age was 47.4 (standard deviation 4.5 years, range 34.5 to 60).”

Discussion:

“The weak correlations found between epigenetic and chronological ages (0.022 for blood, 0.115 for buccal cells) should be considered with the knowledge that the standard deviation of age is 0.16 years (range 53-54 years). Horvath¹, using data from across 82 studies, compared the standard deviation of age measured in each study with the correlation coefficient found in each study between epigenetic and chronological age. He found a correlation of 0.49 between the SD of age and the

performance of his epigenetic clock (in terms of correlation). Thus, with such a small age range in our sample, it should be no surprise that we find a diminished correlation. In ALSPAC, by comparison, where the SD of age is larger at 4.5 years, the correlation between epigenetic and chronological age is 0.53. Comparing the median absolute error the difference is much smaller, with 4.1 years in NSHD and 3.9 years in ALSPAC. This suggests that while the correlation metric is not suitable for NSHD, the epigenetic clock itself is valid for blood samples.”

“While the very small range of age at which the epigenetic information was taken in our sample (standard deviation of 0.16 years, range 53-54 years) explains the low correlation, it does not explain the bias when using the buccal samples. The systematic difference found here appears to be related to tissue specificity, since epigenetic age from blood was closer to chronological age than buccal epigenetic age. The Horvath age estimator was developed using publicly available data covering 51 tissue types (including buccal cells) such that tissue specificity should not result in such an underestimate of chronological age. Three sets of publicly available buccal cell DNA methylation data¹²⁻¹⁴ were among those used in the development of the Horvath epigenetic clock, with a reported correlation of 0.9 between chronological and buccal epigenetic age³. However these were from 109 adolescents¹², 30 newborns (i.e. ten pairs of MZ and five pairs of DZ twins between birth and 18 months¹⁴) and ten individuals who were aged 16, 27, 28, 29, 37, 42, 44, 44, 52, 68¹³. The systematic difference (of 10.7 years) may be explained by the lack of overlap in the age at which information from buccal cells was available in our study and those in the training dataset used to derive the epigenetic clock. Our study questions the use of the epigenetic clock for buccal samples in females between 53 and 60-64.”

4. It is not clear from the Methods exactly how the age acceleration in the blood was adjusted based on this Houseman method. Was a model of the age acceleration residuals constructed, where the Houseman cell composition estimates were adjusted for producing the actual age acceleration residuals, or were the cell composition estimates included as covariates in the models of physical capacity along with the other covariates for blood-based age acceleration models?

Thank you for this comment. We have updated our Statistical Analysis section to clarify the use of the Houseman estimated cell counts for the blood samples.

“The Houseman estimated cell counts were included in the regression of epigenetic age on chronological age to get cell count adjusted AA.”

5. Given that the Horvath method is supposed to be a predictor of age regardless of cell type, it's interesting that the results are so different between buccal and blood based DNA measures of age acceleration. One possible reason could be residual confounding. Both blood and buccal samples are treated essentially the same in the analysis, but half of the blood sample are from individuals who subsequently developed cancer, indicating a case-control sampling for this sample subset. If this sampling scheme is not accounted for in the analysis, the results might be due to this selection and not age acceleration. Again, the lack of a complete description of the sampling scheme(s) (point #1) makes it difficult for the reader to assess the appropriateness of the methods applied and the accuracy of the results.

Thank you for this point, which we have included in our Discussion of the accuracy of the Horvath clock in our samples.

“We found a weak correlation between epigenetic age from blood and buccal cells in the same individuals ($r=0.19$). This low correlation may be due to residual confounding. Since the blood samples come from a case-control study within the birth cohort from which the 790 buccal samples were taken, it may be that the poor correlation between tissues is attributable to selection bias.”

6. Also, the replication population description is missing key points. In particular, how were the mothers selected, how was methylation assessed, and from what cells was the DNA assessed (buccal, blood, or some other tissue)? The reader should not have to read other articles to determine these critical pieces of information.

Thank you for this comment. We have now provided a better description of the ALSPAC methylation data:

“The ARIES study is a subsample of ALSPAC which generated DNA methylation for 1000 families who had biological samples available at each of five time points: umbilical cord blood at birth, peripheral blood at age 7 and 17 in children, and peripheral blood during pregnancy and at 18 years’ follow-up for mothers. The 1000 families were randomly selected from those who had full data available. We used the mother’s follow-up data to replicate our analysis because they best reflected the available NSHD women. All DNA methylation wet-lab and pre-processing analyses were performed at the University of Bristol as part of the ARIES project. Following extraction, DNA was bisulphite converted using the Zymo EZ DNA Methylation™ kit (Zymo, Irvine, CA). Infinium HumanMethylation450 BeadChips were used to measure genome-wide DNA methylation levels at over 485,000 CpG sites. The arrays were scanned using an Illumina iScan, with initial quality review using GenomeStudio. The assay detects methylation of cytosine at CpG islands using two site-specific probes – one to detect the methylated (M) locus and one to detect the unmethylated (U) locus. Single-base extension of the probes incorporates a labelled chain-terminating ddNTP, which is then stained with a fluorescence reagent. The ratio of fluorescent signals from the methylated site versus the unmethylated site determines the level of methylation at the locus. The level of methylation is expressed as a “beta” value (β -value), ranging from 0 (no cytosine methylation) to 1 (complete cytosine methylation). β -values are reported as percentages.”

7. On page 8, line 42, what is meant by the term “minimum sex-specific value”? The study was only conducted in females. So, are the authors using the minimum value in the study or some other reference to determine the minimum value in women?

Thank you for spotting this, we have dropped sex-specific from the description.

8. On page 10, line 50, there is an incomplete sentence.

Thank you for finding this. We have dropped this sentence from the manuscript.

9. In general, the abbreviations are left undefined in the Tables, and some are left undefined in the text (eg. “MRC”).

Thank you for the comment. We have changed the title, dropping the MRC acronym and used footnotes in Tables for the acronyms AA, BMI, GCE and SEP.

Reviewer: 3

Reviewer Name: Dr. Faisal I Rezwan

Institution and Country: University of Southampton

Please state any competing interests: None

Please leave your comments for the authors below

This paper focuses on the study to find the association between age acceleration (AA, the difference

between chronological and epigenetic age) and physical capability measures in mainly female subjects aged over 50 years. It also presented an extension by exploring the association between AA and some life measures. Here authors have measured epigenetic age from buccal and blood samples, which is an interesting approach to understand the variation in epigenetic age measures due to different tissue types though Horvath method is independent of tissue-specificities. Authors have identified association between AA and grip strength. However, the direction of association reversed with ~10 years difference. The paper is well written and easy to read (except few places, mentioned below), and existing literature is well explored and cited. However, I have some major concerns that I feel are required to be addressed and this will help the authors to improve the article.

Major comments:

1. I have found the title of the paper quite obscure as it lacks in the reflection of the whole research. The title is very broad and does not represent the study undertaken and explained in the paper.

We have now included a new title:

“Are objective measures of physical capability related to accelerated epigenetic age? Findings from a British birth cohort.”

2. It is understandable that anyone would find the study very much sex-specific as only female subjects were used and authors have rightly addressed this limitation. However, while presenting the result the authors put strong decision based on inconclusive results. For example: Conclusion in Abstract section stated that evidence have been found on AA associated with greater decrease of grip strength. However, this result is for female only. Therefore, though the use female subject only was mentioned in the limitation, in conclusion that must have been emphasised.

Thank you for this comment. We have updated our Abstract and main document conclusions:

Abstract:

“Conclusions: We found evidence that AA in blood is associated with a greater decrease in grip strength in UK females between 53 and 60-64, but no association with standing balance time or chair rise speed was found.”

Discussion:

“In conclusion, having a higher epigenetic than chronological age is associated with a greater decline in grip strength in UK females between 53 and 60-64, but overall there is little evidence that AA is associated with physical capability change in these women. AA does not appear to be related to measures of physical capability in women at ages 53 or 60- 64, while BMI appears to be associated with accelerated epigenetic age in this population.”

3. In the objective, authors mentioned about investigating the association between epigenetic age and physical capability in later life. However, actually investigation on the association between AA, not epigenetic age, and objective measures of physical capabilities was performed. Therefore, authors really need clarify their aims/objectives properly.

Thank you for this comment. We have replaced “epigenetic age” with “epigenetic age acceleration” where appropriate.

4. For socio economic position, it is not clear why authors used paternal occupational class. A normal trend is always to use maternal socio-economic status or educational level. In Method section, it was also not clear whether the socio-economic position was the combination of paternal and

subjects' socio-economic positions or not. However, this has been cleared in the statistical analysis section. I would recommend to clarify this in the Method section.

Thank you for this comment. Paternal occupation was used as it is a useful variable to adjust for given the time when these data were collected (i.e. 1946). Paternal occupation is a widely used marker of childhood SEP in NSHD and other studies. We have updated our Methods section to better reflect how socio-economic class in adulthood was used:

"Mortality risk factor data were available at age 53 on smoking status (current, never or ex-smoker), nurse measured height (cm) and weight (kg) (used to calculate BMI (kg/m²)), and socio-economic position (SEP) in both childhood (father's occupational class) and adulthood (both education and occupational class were available and adjusted for separately)."

5. Authors has mentioned height, BMI, smoking and SEP as mortality risk factors in the Statistical Analysis section. However, it was not explained how they fixed these as mortality risk factors. A reference in this case is required.

We have added references for each of these at the end of our Introduction:

"We also use data from the NSHD to investigate whether increased epigenetic age is associated with mortality risk factors; smoking¹⁵, higher body mass index (BMI)^{16 17} and more disadvantaged socioeconomic position (SEP)¹⁸."

6. In the last paragraph of the Statistical Analysis section (Page 8, Lines 37 -42), authors mentioned about using minimum sex-specific values for imputation. It is not clear what does this mean. I would recommend to use some imputation methods (such as: MICE) for imputation, not just replacing missing values with minimum values.

Thank you for this comment. Using multiple imputation by chained equations is the standard approach when there are variables which are good predictors of the missing variables, and those variables are missing at random. However, in the current study the missing data are due to participants being unable to perform the tests and so there is informative dropout or a missing not at random process in operation. We know therefore that these individuals are at the very bottom of the distribution in terms of the capability measure. It was therefore decided to use the minimum possible in order to be able to include this high risk group in the analyses in a sensitivity analysis. A sex-specific minimum was mentioned in error since this is usually performed across both men and women, while our study has focussed on only females with methylation available. We have clarified this in our methods section:

"For each of the three measures of physical capability, those who were unable to perform each task for health reasons (Table 1 includes percentage unable to perform each task) were allocated the minimum value observed."

7. The correlation between chronological age and epigenetic ages (both from blood and buccal) is very low. Horvath's online clock generally generates error or warning messages, which sometimes worth to check. It looks like the 353 CpG sites, required for calculating epigenetic age, may have low quality methylation values in those samples. While authors put a reasonable discussion why this may happen in case of buccal tissue, epigenetic age from blood samples shows that the age of samples does not really matter if they are adult. The correlation coefficient between epigenetic age and chronological age is still very low and this is alarming.

Thank you for this comment. Indeed the correlation is very low but this is mostly due to the low range of chronological age in our study. In the original Horvath paper he shows that the standard deviation

of age is heavily correlated with the correlation coefficient between chronological and actual age. In our study the SD of age is 0.15 years, while in the Horvath paper this varies from cohort to cohort (he used 82 cohorts), but one of these had a SD of over 50 years. It is recommended to use median error in place of correlation for low SD samples, and we have now emphasised that measure in our revised document:

Results:

“The median absolute error between chronological and epigenetic age is 10.5 and 4.1 years for buccal and blood respectively. Correlation with chronological age was much lower than previously reported: 0.022 ($p=0.79$) for blood age and 0.115 ($p=0.16$) for buccal age but should be considered with the SD of age being 0.16 years here. Correlation was slightly higher between the two epigenetic ages, with a Pearson correlation coefficient of 0.190 ($p=0.02$).”

“The median absolute error and correlation between epigenetic and chronological age was 3.9 years and 0.53 in ALSPAC respectively, where the chronological age was 47.4 (standard deviation 4.5 years, range 34.5 to 60).”

Discussion:

“The weak correlations found between epigenetic and chronological ages (0.022 for blood, 0.115 for buccal cells) should be considered with the knowledge that the standard deviation of age is 0.16 years (range 53-54 years). Horvath¹, using data from across 82 studies, compared the standard deviation of age measured in each study with the correlation coefficient found in each study between epigenetic and chronological age. He found a correlation of 0.49 between the SD of age and the performance of his epigenetic clock (in terms of correlation). Thus, with such a small age range in our sample, it should be no surprise that we find a diminished correlation. In ALSPAC, by comparison, where the SD of age is larger at 4.5 years, the correlation between epigenetic and chronological age is 0.53. Comparing the median absolute error the difference is much smaller, with 4.1 years in NSHD and 3.9 years in ALSPAC. This suggests that while the correlation metric is not suitable for NSHD, the epigenetic clock itself is valid for blood samples.”

8. It was not clear whether authors checked for outliers and discarded them or not. For mentioned sample sizes, some outliers may really dominate the regression outcome. Therefore, I would recommend the authors to check for outliers, discard them and re-run the analyses.

Thank you for this comment. The data from NSHD are well cleaned and did not present with any outlying observations.

9. As mentioned above, authors have made some strong comments on the finding of the study based on inconclusive results. The results between different age groups are inconsistent and replication cohort could not show the same findings. For example: in Conclusion, authors mentioned that AA is associated with a greater decline of grip strength in middle age and the term “middle age” has been used loosely here.

Thank you for this comment, we have removed the word “middle aged” from our manuscript.

Minor comments:

1. Page 3 Point from Line 21 is incomplete.

Thank you. We have increased the box size so the sentence can be seen.

2. First paragraph of Discussion Page 13 Line 6 -17, is not clearly written. I would recommend rewriting/rephrasing that part specifically from Lines 8 -12.

Thank you for this comment, we have removed the sentence and clarified the paragraph:

“Our study is one of the first to examine epigenetic age from different tissues on the same individuals in relation to risk factors for mortality. We used serial measures of physical capability on the same individuals over time, allowing for better inferences on changes in physical capability in late midlife, compared with having just cross sectional data.”

3. Page 14 Line 16, unit of grip strength “kg” is missing.

Thank you for spotting this, we have included the missing kg

References

1. Starnawska A, Tan Q, Lenart A, et al. Blood DNA methylation age is not associated with cognitive functioning in middle-aged monozygotic twins. *Neurobiol Aging* 2017;50:60-63.
2. Belsky DW, Moffitt TE, Cohen AA, et al. Telomere, epigenetic clock, and biomarker-composite quantifications of biological aging: Do they measure the same thing? *bioRxiv* 2016:071373.
3. Horvath S. DNA methylation age of human tissues and cell types. *Genome Biol* 2013;14(10):R115.
4. Rousseau K, Vinall L, Butterworth S, et al. MUC7 haplotype analysis: results from a longitudinal birth cohort support protective effect of the MUC7* 5 allele on respiratory function. *Ann Hum Genet* 2006;70(4):417-27.
5. Kuh D, Pierce M, Adams J, et al. Cohort profile: updating the cohort profile for the MRC National Survey of Health and Development: a new clinic-based data collection for ageing research. *Int J Epidemiol* 2011;40(1):e1-e9.
6. Teschendorff AE, Yang Z, Wong A, et al. Correlation of Smoking-Associated DNA Methylation Changes in Buccal Cells With DNA Methylation Changes in Epithelial Cancer. *JAMA oncology* 2015;1(4):476-85. doi: 10.1001/jamaoncol.2015.1053 [published Online First: 2015/07/17]
7. Anjum S, Fourkala E-O, Zikan M, et al. A BRCA1-mutation associated DNA methylation signature in blood cells predicts sporadic breast cancer incidence and survival. *Genome Med* 2014;6(6):47.
8. Aryee MJ, Jaffe AE, Corrada-Bravo H, et al. Minfi: a flexible and comprehensive Bioconductor package for the analysis of Infinium DNA methylation microarrays. *Bioinformatics* 2014;30(10):1363-69.
9. Troyanskaya O, Cantor M, Sherlock G, et al. Missing value estimation methods for DNA microarrays. *Bioinformatics* 2001;17(6):520-25.
10. Maksimovic J, Gordon L, Oshlack A. SWAN: Subset-quantile within array normalization for illumina infinium HumanMethylation450 BeadChips. *Genome Biol* 2012;13(6):R44.
11. Teschendorff AE, Marabita F, Lechner M, et al. A beta-mixture quantile normalization method for correcting probe design bias in Illumina Infinium 450 k DNA methylation data. *Bioinformatics* 2013;29(2):189-96.
12. Essex MJ, Thomas Boyce W, Hertzman C, et al. Epigenetic vestiges of early developmental adversity: childhood stress exposure and DNA methylation in adolescence. *Child Dev* 2013;84(1):58-75.
13. Rakyán VK, Down TA, Maslau S, et al. Human aging-associated DNA hypermethylation occurs preferentially at bivalent chromatin domains. *Genome Res* 2010;20(4):434-39.
14. Martino D, Loke YJ, Gordon L, et al. Longitudinal, genome-scale analysis of DNA methylation in twins from birth to 18 months of age reveals rapid epigenetic change in early life and pair-specific effects of discordance. *Genome Biol* 2013;14(5):R42.
15. Doll R, Peto R, Boreham J, et al. Mortality in relation to smoking: 50 years' observations on male British doctors. *BMJ* 2004;328(7455):1519.

16. Lawlor DA, Hart CL, Hole DJ, et al. Reverse causality and confounding and the associations of overweight and obesity with mortality. *Obesity* 2006;14(12):2294-304.
17. Smith GD, Hart C, Upton M, et al. Height and risk of death among men and women: aetiological implications of associations with cardiorespiratory disease and cancer mortality. *J Epidemiol Community Health* 2000;54(2):97-103.
18. Smith GD, Hart C, Blane D, et al. Lifetime socioeconomic position and mortality: prospective observational study. *BMJ* 1997;314(7080):547.

VERSION 2 – REVIEW

REVIEWER	Christian M Page Oslo centre for biostatistics and epidemiology Oslo University Hospital Oslo, Norway
REVIEW RETURNED	16-Jun-2017

GENERAL COMMENTS	The authors have sufficiently addressed the issues raised by the reviewers. I have no further comments or reservations concerning the manuscript.
---

REVIEWER	Dr. Faisal I. Rezwan University of Southampton, UK
REVIEW RETURNED	01-Jul-2017

GENERAL COMMENTS	Authors have properly addressed all the previous comments appropriately. Therefore, I have no further comments.
---